# Metallic Artifacts’ Reduction in Microtomography Using the Bone- and Soft-Tissue Decomposition Method

**DOI:** 10.3390/s24227108

**Published:** 2024-11-05

**Authors:** Jan Juszczyk, Jakub Pałachniak, Ewa Piętka

**Affiliations:** Faculty of Biomedical Engineering, Silesian University of Technology, 41-800 Zabrze, Poland

**Keywords:** computed tomography, decomposition, microtomography, metallic artifacts

## Abstract

Artifacts in computed tomography and X-ray microtomography are image distortions caused by various factors. Some can be reduced before or during the examination, while others are removed algorithmically after image acquisition. The latter group includes metallic artifacts caused by metal objects in the sample. This paper proposes a new method for eliminating metallic artifacts, applying a bone- and soft-tissue decomposition (BSTD) algorithm to microtomography raw data before the reconstruction process. We show that the decomposition algorithm can effectively remove metallic artifacts in microCT images, which increases the image contrast and allows for better visualization of regions near the metallic elements. For quantity analysis, we computed SSIM and PSNR factors, and we observed values increasing from 0.97 to 0.99 and from 40 dB to 43 dB, respectively.

## 1. Introduction

Computed tomography (CT) is a dynamically developing medical imaging technique. Despite competing imaging methods, such as magnetic resonance imaging (MRI), which was supposed to replace CT since its introduction in the 1980s, it remains the most popular imaging method in radiology to this day [1]. One type of CT is the so-called microtomography. This technique allows for resolution up to 1000 times greater than in classic tomography. The resolution of three-dimensional reconstructions can reach below 1 μm [2,3,4]. Still, the algorithms used for reconstruction rely on the same ideas, and the artifacts appearing in the image are essentially the same.

Beam hardening artifacts are a family of image disruptions observed in CT [5]. Photons are scattered and absorbed as the radiation beam passes through the object. Photons in CT interact with the material in two main ways—through the photoelectric effect and the Compton effect [6]. The overall probability of absorbing a photon in CT is proportional to ${\left(Z/E\right)}^{3}$, where *Z* is the atomic number of the element, and E is the photon energy. The Compton effect does not depend on *Z*, and its probability decreases slightly in photons with higher energy [7]. Hence, high-density materials absorb radiation beams strongly during CT, showing up as distortions in the image [8]. Due to the high atomic number, *Z*, of metals, they absorb much of the radiation, causing artifacts in reconstruction [5,9,10]. These artifacts look like stars or flashes of light. These artifacts occur in patients who have fillings, implants, plates, or screws [10]. A metal material made of elements with high atomic numbers absorbs X-rays significantly, so not enough radiation reaches the detector. When reconstructing the image, the algorithm incorrectly calculates the voxel intensities, manifesting in image distortions. These artifacts significantly deteriorate image quality and may make areas essential for the diagnosis no longer visible. This applies to areas near the metal element in particular [10].

The simplest solution to photon absorption is to use a higher beam power by increasing the charge or exposure time. However, this approach only slightly reduces artifacts and increases the patient’s radiation dose [11]. In recent years, there has been growing interest in LDCT—low-dose CT—which places even greater demands on artifact removal algorithms [12]. Various MAR (metal artifact reduction) algorithms have been proposed to reduce artifacts without increasing the radiation dose [9]. Currently in use are MARs based on backprojection, such as single-energy MAR from Toshiba, MAAR for orthopedic implants from Philips, the iterative MAR "iMAR" from Siemens, and Smart MAR from GE Healthcare [7,10,11,13].

MAR based on backprojection involves segmenting the distorted part of the image (metal implant) and replacing it with correct values. The segmentation can occur both in projections (projection-based metal segmentation) or the resulting image (image-based metal segmentation). In the case of segmentation in the image, the implant is identified, and then its location is transferred to projections where the erroneous signal [14] is eliminated. It is the most popular approach used in current methods [7]. This algorithm can be described in four steps [14]:Segmentation of image pixels containing metal using thresholding;Transfer of the segmented region into a projection (sinogram);Distorted data replacement with the interpolated data;Backprojection (reconstruction) using a corrected sinogram.

MARs based on backprojection are clinically applicable for several reasons. The main advantage is that there is no need to increase the radiation dose delivered to the patient. In addition, the radiologist can decide whether to use the algorithm or not: they have access to the image before and after artifact reduction, so the diagnosis can rely on more data [13]. However, these algorithms may produce new artifacts affecting image quality [7].

Hard- and soft-tissue decomposition, or bone- and soft-tissue decomposition (BSTD), is an operation that involves separating the soft tissue and bones in an X-ray image into two separate images—one showing the soft tissue only and the other the bones. The soft tissue image interpolates the background where the bone was calculated. The other contains the bones only and features a better contrast and greater detail than the original image [15]. This operation is significantly different from segmentation. Segmentation involves creating a mask and transferring the pixels inside the mask to a new image in order to remove the background outside the mask [16]. Figure 1 shows the difference between these two operations.

In this paper, we propose a new approach to reducing beam-hardening artifacts using BSTD in order to eliminate part of the signal associated with tissues with a high absorption coefficient from the input images using image decomposition. In the literature, we have not identified the use of BSTD to reduce metal or hard-tissue artifacts in CT or X-ray microtomography. This study aims to investigate the possibility of using the decomposition of projections into soft and hard tissue (BSTD) to reduce artifacts caused by dense materials and improve the quality of reconstruction in microtomography. The proposed solution improves the quality of reconstruction in microtomography in both situations—when interesting elements of the scanned object are located close to the object, generating artifacts, and when the key element strongly absorbs radiation. Our main contribution presented in this paper is to extend the usability of BSTD, which is a method dedicated to two-dimensional images, to three-dimensional reconstruction and to show how this method can be used in this case.

## 2. Materials and Methods

Microtomography imaging uses a flat panel detector that records planar 2D images. For this reason, each projection registered via the detector can be assumed to be an independent X-ray image, so the BSTD can be applied directly. However, we want to introduce the fundamentals of the BSTD for better clarification in our approach.

### 2.1. Soft-Tissue Imaging

Under the assumption that X-rays are the only source of photons, an X-ray image can be presented using the equation [15]
(1)f(x,y)=U(x,y)(1−S(x,y))+S(x,y)

By denoting the obtained X-ray image as f(x,y), we model the resulting image as a combination of a soft-tissue component, S(x,y), and a component of bones or objects with a higher density, U(x,y). The multiplication of U(x,y) by (1−S(x,y)) is used to modulate the bone image with soft tissues. When the transmittance S(x,y) is high (low soft-tissue absorption), (1−S(x,y)) is low, which means the bone-image contribution, U(x,y), to the final image is reduced. Thus, when soft tissues are more transparent to X-rays (lower S(x,y)), the bone image is more visible [17].

If the binary mask containing the hard-tissue area is denoted as M(x,y) (Equation (Equation 2)), the original image containing both soft and hard tissues as f(x,y), and the processed soft tissue image in which the dense elements have been replaced with soft tissue as S(x,y), the problem of determining the pixel intensity in the image S(x,y) can be described by the following equation: (2)$$min{\;\int \int \;}_{M}{∥\nabla S∥}^{2},\;\mathrm{where}\;S_{\partial M}=f_{\partial M},$$
where ∂M denotes the mask boundaries. The goal is to find a function, S(x,y), that minimizes the sum of squared gradients in the area covered by the mask M(x,y). The gradient ∇S, which is a vector of partial derivatives of S(x,y) concerning *x* and *y*, measures local changes in pixel intensity. Its norm $∥\nabla S∥$ measures the steepness of the changes, and its square highlights areas with rapid changes in intensity that we want to avoid, promoting smooth solutions. In other words, we are looking for the smoothest possible image (i.e., without intensity changes), which simultaneously corresponds to the original image on the mask borders.

The boundary condition $S_{\partial M}=f_{\partial M}$ ensures that the image-*S* intensities at the mask boundaries ∂M exactly match the values in the original image, *f*. This is key to securing the continuity and naturalness of the image at its edges, eliminating artificial or unnatural transitions between processed and unprocessed areas. The reduction of the problem to Laplace’s equation,
(3)$$\Delta S=0,\;\mathrm{where}\;S_{\partial M}=f_{\partial M}$$
comes from the requirement that the *S* function should be as smooth as possible inside the mask. Laplace’s equation, ΔS=0, where the Laplacian ΔS is the sum of second partial derivatives, suggests that *S* contains no local extrema inside the mask, which is equivalent to no internal sources of intensity changes. Due to the fact that soft tissue has a uniform structure in X-rays, that is a good way to estimate soft tissue in medical imaging. Equation (Equation 3) can be solved numerically using the convolution pyramid method, described in detail in [15,17].

### 2.2. Convolution Pyramids

Laplace’s equation and its applications have been used for seamless image cloning and combining images without visible edges for many years [18,19]. They were used to create membranes—masks describing interpolation within a given region of interest (ROI) [17]. The convolution pyramid method is based on the Shepard method–a multidimensional approximation approach for distributed sets of points [20]. It identifies an ROI whose boundaries are b(x) and then calculates a weighted average [17]:(4)$$r\left(x\right)=\frac{\sum_{k}w_{k}\left(x\right)b\left(x_{k}\right)}{\sum_{k}w_{k}\left(x\right)},$$
where $w_{k}$ is a weight function that determines how much the point on the border influences the interpolated pixel. It is empirically defined as follows:(5)$$w_{k}\left(x\right)=w\left(x_{k},x\right)=\frac{1}{d{\left(x_{k},x\right)}^{3}},$$
where $d(x_{k},x)$ represents the distance between the point $x_{k}$ and the point *x*, where $x_{k}$ is a boundary point, and *x* is any point inside the ROI. It is high within the image boundary and drops rapidly while moving away from the border. The original form of Equation (Equation 4) features high computational complexity. With *K* boundary points in an ROI of *n* points, the computational complexity is O(Kn). The proposed method allows calculations in O(n) time. The first step is to expand *b*, such that the value is equal to the value of the input image at the image boundary, while it is zero otherwise:(6)$$\hat{r}\left(x_{i}\right)=\left\{\begin{array}{cc} b(x_{k}) & \mathrm{for}\;x_{i}=x_{k}\;\mathrm{on}\;\mathrm{the}\;\mathrm{boundary},\\ 0 & \mathrm{otherwise}. \end{array}\right.$$

Then, Equation (Equation 4) can be described as a ratio of convolutions:(7)$$r\left(x_{j}\right)=\frac{\sum_{i=0}^{n}w\left(x_{j},x_{i}\right)\hat{r}\left(x_{i}\right)}{\sum_{k=0}^{n}w\left(x_{j},x_{i}\right)\chi_{\hat{r}}\left(x_{k}\right)}=\frac{w*\hat{r}}{w*\chi_{\hat{r}}},$$
where $\chi_{\hat{r}}$ is a characteristic function of *r*. Including χ in the denominator ensures that the weights assigned to zeros in *r* (that is, to the areas inside the boundaries irrelevant to the interpolation of the boundary values) are not considered in the final result. That makes the interpolation effect more consistent with the actual values at the borders, and the areas that should not affect the interpolation are eliminated. The optimal size of kernels used for convolutions was empirically determined in [17].

Convolution kernels process different image resolutions to analyze and synthesize images gradually. In this method, the input signal *a* is first processed via the filter $h_{1}$:(8)$$a^{l+1}=↓\left(h_{1}*a^{l}\right).$$

The *l*th level of the signal $a^{l}$ is filtered via $h_{1}$ and then subsampled (↓), which reduces its resolution by half, preparing it for further processing at the next level.

Synthesis starts from the lowest resolution level and involves the gradual reconstruction of image details at increasing levels:(9)$$a^{l}=h_{2}*\left(↑a^{l+1}\right)+g*a_{0}^{l},$$
$a^{l+1}$ is first upsampled to a higher resolution (↑) and then filtered by $h_{2}$. Additionally, at each level, the result of the *g* filter operation on the original data from that level $a_{0}^{l}$ is added to the signal. That allows for the reconstruction and integration of details at all pyramid levels.

Using the input image and assuming that the convolutional pyramids’ outcome is a good approximation of the soft-tissue image, we can perform the decomposition. It is an optional step in the algorithm, but it has great potential to replace two- and three-dimensional segmentation algorithms. This decomposition aims to separate the original X-ray image, f(x,y), into an estimated soft-tissue image, S(x,y), and bone image, U(x,y). S(x,y) can be considered an image with a reduced influence of dense tissues.

### 2.3. The Method

In microtomography, the decomposition method separates projection images into soft- and hard-tissue components. Then, the impact of beam hardening on soft-tissue reconstruction can be significantly reduced using the soft-tissue image to perform the reconstruction. The hard-tissue image can be reconstructed separately, and then, as the method results, a pair of soft- and hard-reconstruction images can comprised. The proposed method (Figure 2) consists of the following operations:

Acquisition projections $f_{i}\left(x,y\right)$ (2D).Calculation of the mask $M_{i}\left(x,y\right)$ for each projection.Decomposition for each projection ($S_{i}\left(x,y\right)$ and $U_{i}\left(x,y\right)$).Reconstruction based on $S_{i}\left(x,y\right)$ and $U_{i}\left(x,y\right)$ using FDK algorithm [21].

**Figure 2 sensors-24-07108-f002:**
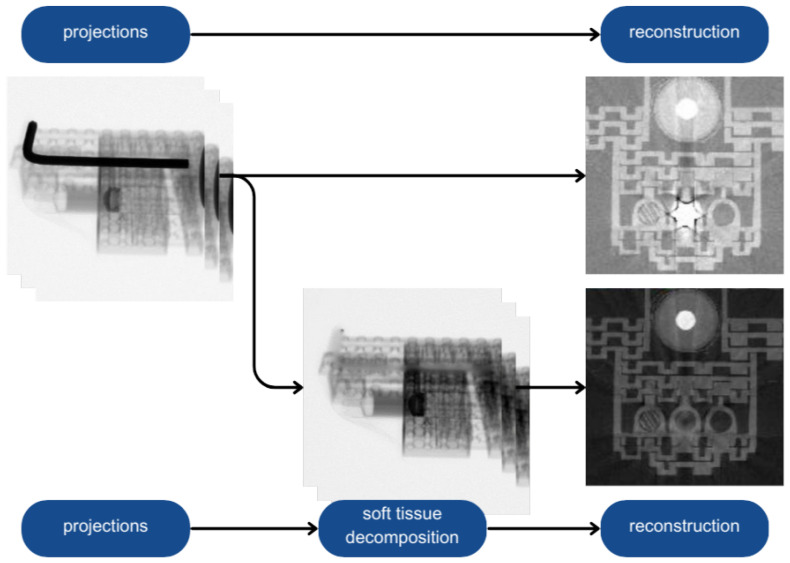
General workflow of the method. A diagram of obtaining tomographic images without artifact reduction (**top**) and with the decomposition stage (**bottom**).

The mask is the basis for determining the location of the element to be removed and replaced with the soft-tissue model. As it does not have to adhere precisely to the removed object boundary, its determination is possible in many ways, e.g., thresholding, morphological operations, or manual delineation. It is also possible to use machine learning methods. The impact of the precise boundary delineation will be discussed in Section 3. The soft-tissue image constitutes input data for the reconstructor. By setting an appropriate masking threshold, we can remove only parts of the image related to BH artifacts. Therefore, the reconstructor analyzes projections without hard objects. The decomposition of sample data (a phantom with a simulated metal implant) is shown in Figure 3.

## 3. Results

The method was validated in two experiments. The first test involved an original phantom that allowed the source of artifacts to be placed inside, without changing the geometry. The porcine tissue was analyzed in the second test. The microtomography scans were performed using the PXR COMPACT CT device (ProCon X-ray GmbH, Sarstedt, Germany).

### 3.1. Experiment I

For the first experiment, we prepared a phantom made of Lego bricks and a wooden strip representing objects of different densities. Lego-based phantoms are widely used in imaging system testing due to their high quality and easy accessibility [22,23,24]. A metal Allen key simulated an implant. It could be removed from the phantom without changing its geometry or reconstruction and exposure parameters. We performed the experiment twice, with and without a metal element, with a fixed phantom position in the device.

#### 3.1.1. The Impact of Mask Size on Reconstruction Quality

The mask obtained via thresholding was used for the image decomposition with six manually selected thresholds. We performed the reconstruction using the FDK algorithm [21] with a Hamming filter for each of the six cases. Figure 4 shows sample results of all six runs with a normalized brightness (zoomed-in area). Note that the normalization lowers the contrast in illustrations where the mask does not cover the whole metal element.

The boundaries of the implant in Figure 4, together with the simultaneous decrease in the brightness of the middle, indicate that the mask did not cover the entire implant, leaving pixels that create reconstruction artifacts and distort the image near the metal element. The contrast of the image and detail of the area improve with an increasing threshold. The artifacts also vanish, some image details are restored, and the whole image is easier to interpret.

The decomposition increased the average intensity of pixels outside the masked area. The sudden increase in intensity in the middle of the profile was successfully eliminated, and data regarding sleeve boundaries near the metal element were recovered.

#### 3.1.2. Quantitative Analysis

With the top reconstruction quality achieved using the threshold of 10,000, we performed the following comparative analysis in this setup. First, we normalized the pixel intensities to the level of the image without the implant (reference image) by equalizing the background level (both with and without MAR). Sample results are shown in Figure 5. The artifacts visible in the MAR-less image (Figure 5a) are reduced after applying MAR (Figure 5c). Moreover, after using MAR, the contrast is improved.

To better illustrate the differences between the reconstruction with and without MAR, we determined maps of the absolute differences in pixel brightness between both reconstructed images and the reference image. Figure 6. shows maps of the absolute differences in pixel brightness, and the reference image is shown in Figure 5b. They clearly indicate that applying MAR significantly reduces reconstruction errors compared to the MAR-less approach. Without MAR, the artifacts occur not only near their source but also in the background area.

Figure 7 shows profiles of pixel-intensity changes in two sample locations marked with white horizontal lines. The top profile (Figure 7d) does not cross the implant. The intensities in the MAR profile are closer to the reference. The contrast is higher, too. In the bottom profile (Figure 7e), the profile region of interest is located between the 350th and 580th columns. The artifacts in the image without MAR do not allow for indicating the sleeve boundary. Instead, it is visible after the reconstruction with MAR is applied.

To quantify the impact of MAR on reconstruction quality, we employed two measures. First, the structural similarity index (*SSIM*) is given by the following:(10)$$SSIM=\frac{\left(2\mu_{x}\mu_{y}+c_{1}\right)\left(2\sigma_{xy}+c_{2}\right)}{\left(\mu_{x}^{2}+\mu_{y}^{2}+c_{1}\right)\left(\sigma_{x}^{2}+\sigma_{y}^{2}+c_{2}\right)},$$
where

$\mu_{A}$, $\mu_{B}$—mean value of pixels of image A and image B;$\sigma_{A}$, $\sigma_{B}$—variance of pixels of image A and image B;$\sigma_{AB}$—covariance of pixels of images A and B;$c_{1}$, $c_{2}$—constants [25].

Second, the peak signal-to-noise ratio (*PSNR*) is given by
(11)$$PSNR=20\cdot \log_{10}\left(MAX_{I}\right)-10\cdot \log_{10}\left(MSE\right),$$
where $MAX_{I}$ is the maximum pixel intensity in the original image, and *MSE* is the mean squared error. The results are given in Table 1.

We have also compared our method with three methods: NMAR [26] and LI-MAR [27], which are similar in idea to our method and are widely used as a reference method [28,29,30], and BHC Streaking, developed by Mitos (MITOS GmbH, München, Germany) and delivered with X-Aid software (ver. 9999.9.9). The results are given in Table 2. We decided to compare projections to make the results independent of the reconstruction process.

Figure 8 shows the intensity profiles and corresponding intensities for different methods. This comparison allows us to evaluate the effectiveness of our approach in relation to established techniques in the field.

Each MAR method based on interpolation before reconstruction provides significantly better results than BHC. They greatly enhance the contrast and ensure a similar level of intensity across the entire profile, except for the very center. Our proposed method reconstructs areas of lower intensity in the central part of the profile, where the implant was located, while LIMAR and NMAR provide a more uniform result. Despite NMAR’s higher accuracy than LIMAR on natural data sets [26], in our experiment, all three methods dealt with the artifact in a very similar manner. However, the expected result in the place where the artifact source was removed is empty space, which is better provided via our method.

### 3.2. Experiment II

Image decomposition allows for obtaining images of soft and hard tissues. In addition to using decomposition to reduce metal artifacts, it is possible to use the image of hard tissues to improve the visualization of bones. In Experiment II, we scanned a piece of pork with rib bones and performed deconvolution of the obtained projections. In this case, we used the image of hard tissues at further processing stages. As a result, we eliminated soft tissue from the images of bones (Figure 9).

Performing reconstruction on the bone image resulted in a lower average pixel intensity in the image (Figure 9c). The contrast increased, which is a desirable effect (Figure 9d). The operation allowed for the imaging of porous structures that might not have been visible without soft tissue decomposition.

## 4. Discussion

Decomposition of different-density objects in CT or microtomography is an operation that, when used appropriately, can effectively remove high-density artifacts. The deconvolution pyramid method requires a mask defining the area containing dense objects, but it does not specify how the mask should be obtained. In the presented experiments, the thresholding was sufficiently robust for segmentation due to the significant difference in pixel intensity between the object (implant and bone) and the background. More advanced segmentation methods can be used if necessary to determine the correct mask for the proper artifact removal (see Figure 4 and Figure 5). A too-small mask will not remove artifact-related pixels from the source image and lower the algorithm’s efficiency. In Experiment I, it was shown that using MAR for decomposition increases the image contrast, regardless of the distance from the artifact, by up to 45% and significantly improves the visibility of details. The edges of objects around the source of artifacts became substantially more apparent. Therefore, in specific cases, the algorithm can restore image components directly near the metal element, which may be a critical aspect in diagnosing certain diseases, e.g., through better imaging of the bone area near the implant. Moreover, the method allows for the removal of objects that potentially cause artifacts before the reconstruction stage. The intensity profile of our method Figure 8 demonstrates a unique charateristic in that it reconstructs areas of lower intensity in the central part of the profile where the implant was located. This could be advantageous in certain clinical scenarios where differentiation of tissues near implants is crucial.

Experiment II showed that performing reconstruction using the (U) hard-tissue image improves its contrast—a reasonably expected conclusion, given the results presented in [15]. It is a direct extension of this method for reconstruction. There are no restrictions on reconstructing both the hard-tissue image (U) and soft-tissue image (S). In such a case, we obtain two reconstructions, which could increase the clarity of the information. An essential advantage of our method is that it is independent of the reconstruction approach itself—it is possible to use any available algorithm. We obtained the results presented in this paper using the FDK algorithm, but there are no restrictions to using methods from the Algebraic Reconstruction Techniques (ARTs) group. Based on the results of Experiment II, we hypothesize that the algorithm also has the potential to be used for removing the source of noise when the signal associated with very soft tissues is treated as noise, e.g., by covering the entire object with the *M* mask, excluding backgrounds.

The main limitation of our method is the strong dependence on the mask of the artifact source. That implies that we must know the location of the artifact source before the reconstruction process. We also need to determine the artifact source’s mask for each projection, which can be time-consuming. The solution to this issue is to reconstruct the image without MAR, create the mask of the artifact source, and then re-create the masking projections. However, it can still be hard to create the proper mask because of the metal artifacts. Next, the specific feature of our method is that the source of the artifact is completely removed from the image, not only artifacts. However, if the reconstruction of the metallic object is still needed, we can use the “hard tissue projection” for reconstruction.

## 5. Conclusions

The presented method effectively reduces artifacts from metallic objects, but that is not the only application of this method. We assume that this method can improve image quality by dividing the image signal (information) into two parts—soft and hard tissue signals. We hypothesize that presenting two images can provide better detail recognition. Our method was designed for microtomography application; however, it can easily be transferred to another tomographic method, especially cone-beam computed tomography (CBCT).

## Figures and Tables

**Figure 1 sensors-24-07108-f001:**
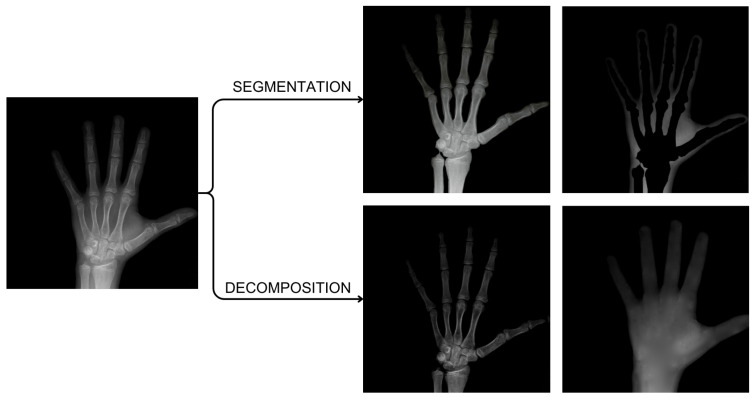
Segmentation vs. decomposition [15].

**Figure 3 sensors-24-07108-f003:**
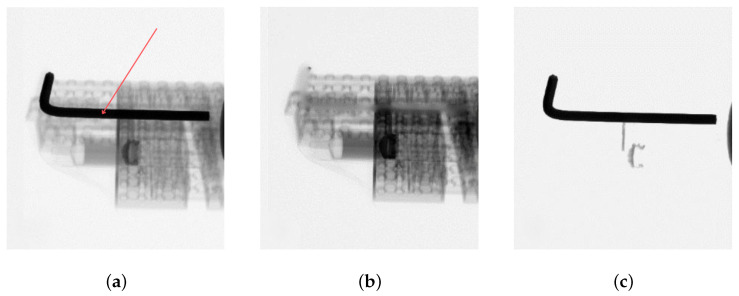
Illustration of the decomposition results of a phantom image with a simulated metal implant: (**a**) original image in which the red arrow points to the metal element, (**b**) decomposed image with the implant removed, (**c**) implant image without the background.

**Figure 4 sensors-24-07108-f004:**
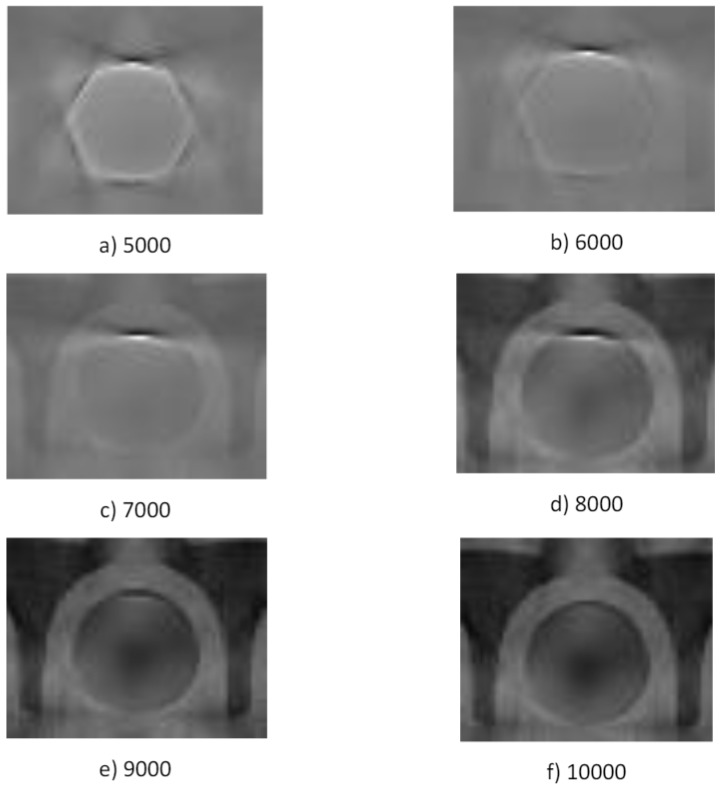
Illustration of the metal implant region (magnification) decomposed with six different thresholds: 5000 (**a**), 6000 (**b**), 7000 (**c**), 8000 (**d**), 9000 (**e**), and 10,000 (**f**).

**Figure 5 sensors-24-07108-f005:**
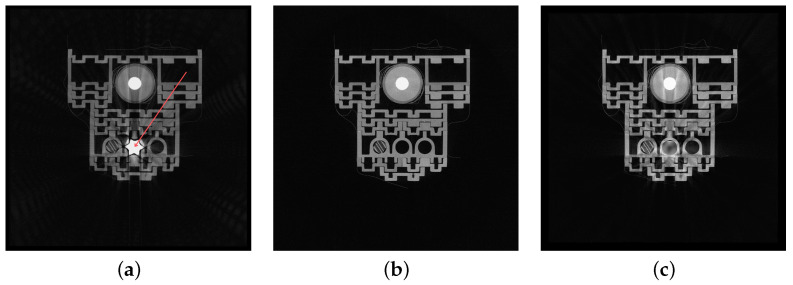
Reconstruction outcome without applying MAR (**a**) (reconstruction without MAR in which the red arrow points to the metal element) without implant—reference image (**b**) reference reconstruction without implant and with MAR (**c**) (reference reconstruction—without implant).

**Figure 6 sensors-24-07108-f006:**
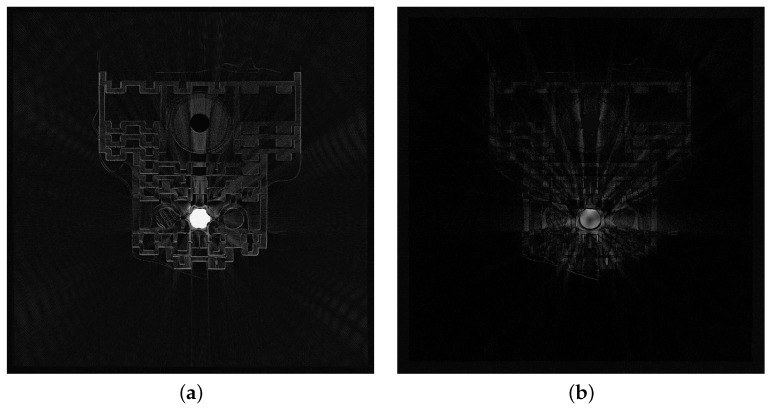
Maps of absolute reconstruction differences regarding reference image: without MAR (**a**) and with MAR (**b**).

**Figure 7 sensors-24-07108-f007:**
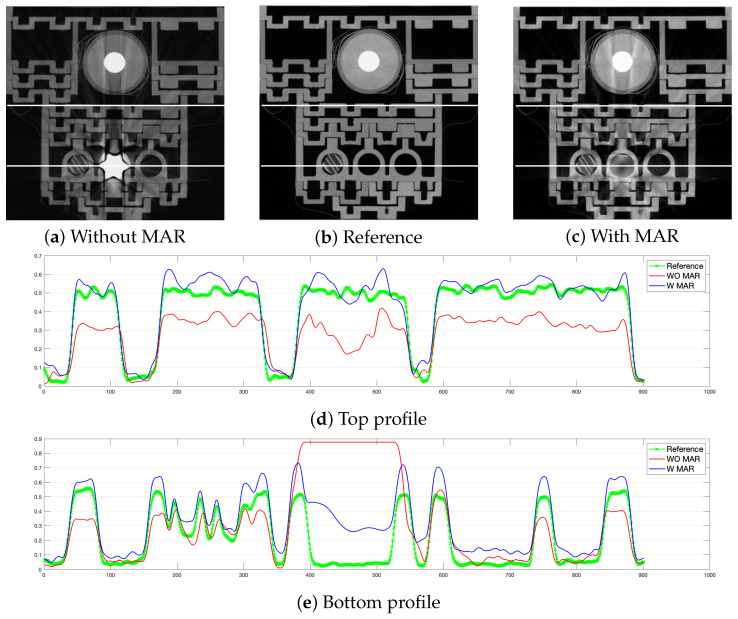
Intensity profiles (**d**,**e**) over image reconstructed without MAR (**a**), reference image, and (**b**) image reconstructed with MAR (**c**).

**Figure 8 sensors-24-07108-f008:**
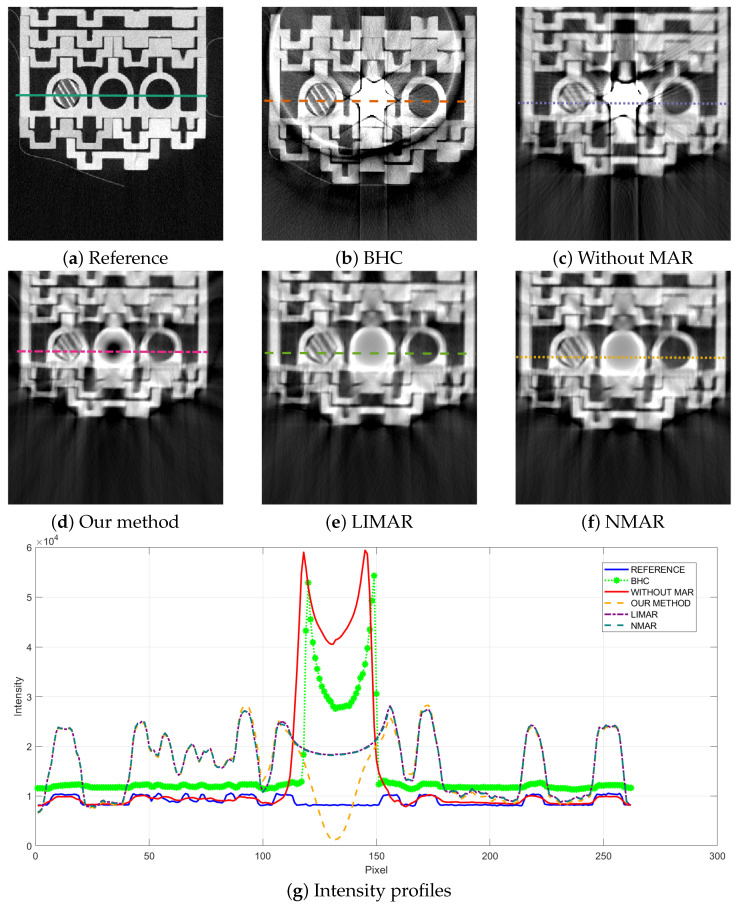
Reconstructions with profile lines (**a**–**f**) and corresponding intensity profiles (**g**).

**Figure 9 sensors-24-07108-f009:**
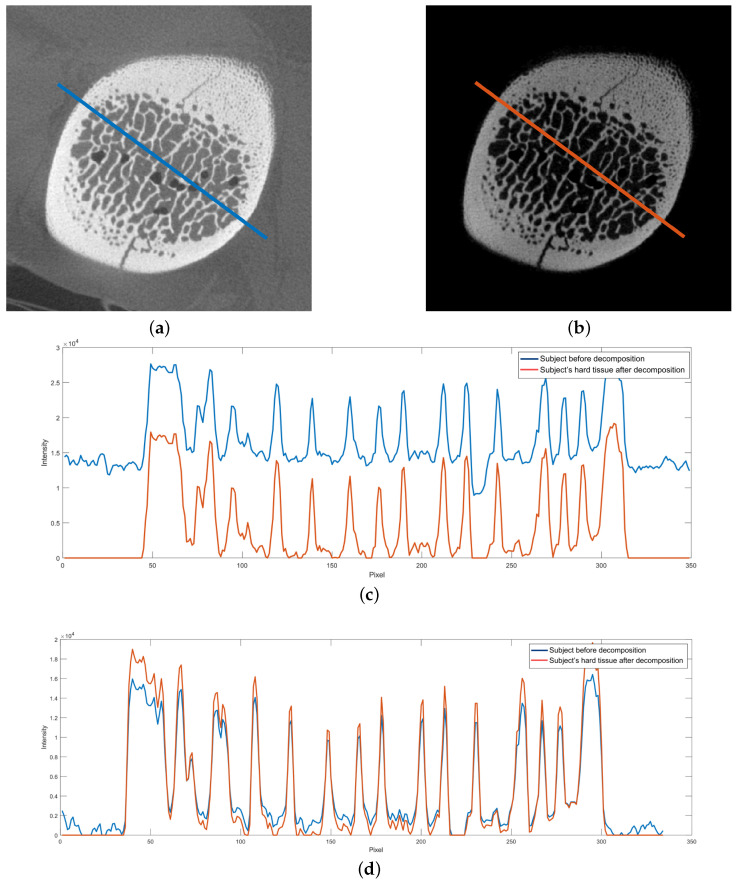
Decomposition applied to improve contrast in bone image. (**a**) Original reconstruction (without MAR). (**b**) Reconstruction of the hard tissue after decomposition. (**c**) Intensity profile: blue—without MAR; red—with MAR. (**d**) Intensity profile after the baseline was removed: blue—without MAR; red—with MAR.

**Table 1 sensors-24-07108-t001:** Reconstruction assessment with and without MAR.

	SSIM	PSNR
Original vs. reconstruction without MAR	0.97	40 dB
Original vs. reconstruction with MAR	0.99	43 dB

**Table 2 sensors-24-07108-t002:** Quality of projections processed with LIMAR, NMAR, and our approach.

	SSIM	PSNR
LIMAR	0.95	32 dB
NMAR	0.95	32 dB
OURS	0.95	31 dB

## Data Availability

Due to the size of the original data set, the sample of the original data presented in the study is openly available in Juszczyk, J., Pałachniak, J. and Pietka, E. (2024), “Metal artifact reduction based on bone and soft tissue decomposition for the microtomography investigation–raw data and reconstructions”, Mendeley Data, V1, doi: 10.17632/txy98jt7yp.1. Please contact the first author for the download availability of the whole data set.

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
