# Peer review of "Metallic Artifacts’ Reduction in Microtomography Using the Bone- and Soft-Tissue Decomposition Method"

_sensors, 2024, doi:10.3390/s24227108_

Round 1

Reviewer 1 Report

Comments and Suggestions for Authors

The authors Juszczyk et al presents an interesting study on the metallic artifacts reduction (MAR) using a proposed algorithm known as a bone and soft tissue decomposition (BSTD). The paper effectively articulates the proposed theory and algorithm, evaluating it through experiments on two phantom samples. The results demonstrate that the original image can be successfully decomposed into two distinct images: one with higher absorption and another with lower absorption. The discussion is clear and provides adequate evidence to support their claims. In my view, this work contributes to the field of micro-tomography artifacts removal algorithms and should be published if the following questions and concerns are addressed:

Given the importance of the metal component in this study, it would be beneficial for the authors to highlight the metal parts in their radiographic images or tomographic slices. Additionally, the legends in Figures 7 and 8 are too small and may hinder clarity.

The paper would benefit from a dedicated conclusion section that summarizes the main results and findings.

There is a lack of comparison between the authors' proposed method and other metal artifacts reduction software. A comprehensive comparison with other widely used commercial MAR methods, such as those from Toshiba, Philips, Siemens, and GE, would strengthen the study.

In summary, I would recommend this paper for publication in Sensors after a thorough revision that addresses the outlined concerns and includes a comparison with existing methods.

Author Response

C1

The authors Juszczyk et al presents an interesting study on the metallic artifacts reduction (MAR) using a proposed algorithm known as a bone and soft tissue decomposition (BSTD). The paper effectively articulates the proposed theory and algorithm, evaluating it through experiments on two phantom samples. The results demonstrate that the original image can be successfully decomposed into two distinct images: one with higher absorption and another with lower absorption. The discussion is clear and provides adequate evidence to support their claims. In my view, this work contributes to the field of micro-tomography artifacts removal algorithms and should be published if the following questions and concerns are addressed:

R1

Thank you for your comment. We addressed all suggested corrections in the text.

C2

Given the importance of the metal component in this study, it would be beneficial for the authors to highlight the metal parts in their radiographic images or tomographic slices. Additionally, the legends in Figures 7 and 8 are too small and may hinder clarity.

R2

We enlarge legends in Figures 7 and 8. and add the red arrows pointing to the metal element.

C3

The paper would benefit from a dedicated conclusion section that summarizes the main results and findings.

R3

We have added section Conclusion with following paragraph:

The presented method effectively reduces artifacts from metallic objects, but that is not the only application of this method. We assume this method can improve image quality by dividing the image signal (information) into two parts - soft and hard tissue signals. We hypothesize that presenting two images can provide better detail recognition. Our method was designed for microtomography application; however, it can be easily transferred to another tomographic method, especially cone beam computed tomography (CBCT).

C4

There is a lack of comparison between the authors' proposed method and other metal artifacts reduction software. A comprehensive comparison with other widely used commercial MAR methods, such as those from Toshiba, Philips, Siemens, and GE, would strengthen the study.

R4

Thank you for your suggestions. We have compared our method with the following methods: NMAR, LI-MAR and BHC Streaking developed by MItos.de and delivered with X-Aid software. The results of this comparison are presented in the table, as well as representative slices and profiles. We have also added a new paragraph in the discussion with comments on the results of this comparison.

C5

In summary, I would recommend this paper for publication in Sensors after a thorough revision that addresses the outlined concerns and includes a comparison with existing methods.

R5

Thank you for this recommendation.

Reviewer 2 Report

Comments and Suggestions for Authors

- The paper is written in a clear way and the content is of interest for the research community.

- Authors didn't compare between their work and state of the art models . Such comparison is crucial in such research paper to evaluate where does the paper stands comparing to other recent methods . The comparison table needs to be added.

- The method over all lacks of novelty as all components have been widely used in the literature. 

- This model serves the medical domain so why practically all your figures are outside the medical field. In my opinion you should highlight the efficiency of your methods on computed tomography and X-ray microtomography images. 

- You need to highlight your clear contributions in the introduction.  

- Figures are clear which added to the quality of the paper.

- The fact that authors highlighted a clear discussion section added to the quality of the paper , though authors should add a clear section that evaluates objectively their method by emphasizing the weaknesses of their method and how it can be improved . Authors should as well show some examples where the algorithm did fail and discuss why it did fail in their opinion.

Comments on the Quality of English Language

The paper is written in a clear way and the english is good enough . I just detected some typos and some double words exp: that that.

Author Response

C1

- The paper is written in a clear way and the content is of interest for the research community.

R1

Thank you for valuable comment.

C2

- Authors didn't compare between their work and state of the art models . Such comparison is crucial in such research paper to evaluate where does the paper stands comparing to other recent methods . The comparison table needs to be added.

R2

We have compared our method with the following methods: NMAR, LI-MAR and BHC Streaking developed by MItos.de and delivered with X-Aid software.

We have also added a new paragraph in the discussion with comments on the results of this comparison. The results of this comparison are presented in the table, as well as representative slices and profiles.

C3

- The method over all lacks of novelty as all components have been widely used in the literature.

R3

Despite all the components of our method being widely used, we did not find such use of methods (BSTD) for metal artifact reduction. The novelty of the study has been emphasized in response to your remarks, including this one:

In the literature, we have not identified the use of BSTD to reduce metal or hard tissue artifacts in CT or X-ray microtomography.

C4

- This model serves the medical domain so why practically all your figures are outside the medical field. In my opinion you should highlight the efficiency of your methods on computed tomography and X-ray microtomography images.

R4

Thank you for this valuable comment.

We tested our method on phantom models, a common practice. It was important in our experiment to obtain comparable results with and without metallic objects. So, we decided to design a phantom where we could remove metallic elements without changing its geometry. Unfortunately, the side effect of this approach is predominance of non-medical images in results.

To highlight the efficiency of your methods on computed tomography and X-ray microtomography images, we add the comparison of our method with other and pointed advantages of our method. Please see R7.

C5

- You need to highlight your clear contributions in the introduction.

R5

To highlight our contributions in the introduction we add the sentence:

Our main contribution, presented in this paper, is to extend the usability of BSTD, which is a method dedicated to 2-dimensional images, to 3-dimensional reconstruction; and to show how this method can be used in this case.

C6

- Figures are clear which added to the quality of the paper.

R6

Thank you for valuable comment.

C7

- The fact that authors highlighted a clear discussion section added to the quality of the paper , though authors should add a clear section that evaluates objectively their method by emphasizing the weaknesses of their method and how it can be improved . Authors should as well show some examples where the algorithm did fail and discuss why it did fail in their opinion.

R7

We have identified the limitations of our method. A new paragraph has been added to the Discussion section:

The main limitation of our method is the strong dependence on the mask of the artifact source. That implies we must know the location of the artifact source before the reconstruction process. We also need to determine the artifact source's mask for each projection, which can be time-consuming.  The solution to this issue is to reconstruct the image without MAR, then create the mask of the artifact source, and then re-create the masking projections. However, it can still be hard to create the proper mask because of the metal artifacts.

Next, the specific feature of our method is the source of the artifact is completely removed from the image, not only artifacts. However, if the reconstruction of the metallic object is still needed, we can use the "hard tissue projection" for reconstruction.

Reviewer 3 Report

Comments and Suggestions for Authors

The article (Manuscript ID: sensors-3239747) propose a new method to eliminate metal artifacts. The new method applies the BSTD algorithm to microCT data to effectively remove metal artifacts, improve image contrast, increase the SSIM from 0.97 to 0.99, and raise the PSNR from 40 dB to 43 dB. Overall, this review can be accepted after the following comments have been processed.

1. L-53: Phrase correction: "segmented fragment" should be "segmented area" or "segmented region". If it refers to an entire area or part of a segmented image, "fragment" usually refers to a smaller, broken part.

2. L-63-64: The one with soft tissue interpolates the background area in place of the bone calculated. This sentence is somewhat unclear and may need to be rewritten to clarify: The soft tissue image interpolates the background where the bone was calculated.

3. L-82: Typing error: In "so the BSTD can by applied directliy.", "by" should be changed to "be" and "directliy" should be changed to "directly". Please pay attention to the correct spelling of the words.

4. L-89: In "bones or objects with higher density component U(x, y))", U(x, y)) should be U(x, y).

5. L-121: They were used to create membranesmasks describing interpolation within a given region of interest (ROI) [23]. In the text, it can be inferred from the context that Laplace's equation is not explicitly mentioned in plural form. Hence, they should be replaced with it to clearly refer to Laplace's equation. If referring to and its applications, consider explicitly stating "These techniques/methods" or "Laplace's equation and its applications" to avoid ambiguity.

6. L-156: The "U(xy)" in "bone image U(xy)" should be "U(x, y)".

7. L-163-164: "comprise a pair of soft- and hard-reconstruction" is not very accurate. If the meaning is "the resulting image can contain both soft and hard tissue reconstructions", then it can be changed to "The resulting image can contain both soft and hard tissue reconstructions".

8. L-165:If the meaning is “acquisition” or “determination” of the projected image, then “determination” should be changed to “acquisition”.

9. Figure 3, in the sentence "Illustration on the decomposition results", "Illustration on of" should be changed to "Illustration of" or, as described below, "Illustration" followed by a colon and the specific content.

10. L-197: Repetitive words: "... indicate that that the mask did not cover the entire implant…”. Here, "that that" is repeated.

11. L-250: "discriminate" should be "specify" or "prescribe". The meaning here may be "specify" or "prescribe" how to obtain the mask.

12. L-254: When referring to figure numbers, usually in the order in which they appear in the text, they should be read "(see Fig. 4 and 5)". If Figure 4 is mentioned before Figure 5 in the text, the order should be correct. If not, adjust the order of the numbers.

13. L-266-267: In such a case, we obtain two reconstructions, which could increase the readability of the information. " Because "readability" is often used to describe the readability of text. Here, you can change to "clarity".

14. L-274: "M mask" may refer to a mask used to focus on a specific area, please give a detailed description and definition of the "M mask" in the context.

15. In the Discussion part, the results were not fully discussed, including the comparison with other methods, the limitations of the method and the direction of future improvement. Therefore, it is recommended to join the discussion.

Comments on the Quality of English Language

Minor editing of English language required.

Author Response

C1

L-53: Phrase correction: "segmented fragment" should be "segmented area" or "segmented region". If it refers to an entire area or part of a segmented image, "fragment" usually refers to a smaller, broken part.

 R1

Thank you for your comment. We made the suggested corrections in the text.

C2

C2

L-63-64: “The one with soft tissue interpolates the background area in place of the bone calculated.” This sentence is somewhat unclear and may need to be rewritten to clarify: “The soft tissue image interpolates the background where the bone was calculated.”

R2

Thank you for your comment. We made the suggested corrections in the text.

C3

C3

L-82: Typing error: In "so the BSTD can by applied directliy.", "by" should be changed to "be" and "directliy" should be changed to "directly". Please pay attention to the correct spelling of the words.

R3

Thank you for your comment. We made the suggested corrections in the text.

C4

C4

L-89: In "bones or objects with higher density component U(x, y))", U(x, y)) should be U(x, y).

R4

Thank you for your comment. We made the suggested corrections in the text.

C5

C5

L-121: “They were used to create membranes—masks describing interpolation within a given region of interest (ROI) [23].” In the text, it can be inferred from the context that ‘Laplace's equation’ is not explicitly mentioned in plural form. Hence, ‘they’ should be replaced with ‘it’ to clearly refer to ‘Laplace's equation’. If referring to and its applications, consider explicitly stating "These techniques/methods" or "Laplace's equation and its applications" to avoid ambiguity.

R5

Thank you for your comment. We made the suggested corrections in the text.

C6

C6

L-156: The "U(xy)" in "bone image U(xy)" should be "U(x, y)".

R6

Thank you for your comment. We made the suggested corrections in the text.

C7

C7

L-163-164: "comprise a pair of soft- and hard-reconstruction" is not very accurate. If the meaning is "the resulting image can contain both soft and hard tissue reconstructions", then it can be changed to "The resulting image can contain both soft and hard tissue reconstructions".

R7

Thank you for your comment. We made the suggested corrections in the text by changing proper sentence to:

The hard tissue image can be reconstructed separately, and then, as the method results, a pair of soft- and hard-reconstruction images can comprised.

C8

C8

L-165:If the meaning is “acquisition” or “determination” of the projected image, then “determination” should be changed to “acquisition”.

R8

Thank you for your comment. We made the suggested corrections in the text.

C9

C9

Figure 3, in the sentence "Illustration on the decomposition results", "Illustration on of" should be changed to "Illustration of" or, as described below, "Illustration" followed by a colon and the specific content.

R9

Thank you for your comment. We made the suggested corrections in the text.

C10

C10

L-197: Repetitive words: "... indicate that that the mask did not cover the entire implant…”. Here, "that that" is repeated.

R10

Thank you for your comment. We made the suggested corrections in the text.

C11

C11

L-250: "discriminate" should be "specify" or "prescribe". The meaning here may be "specify" or "prescribe" how to obtain the mask.

R11

Thank you for your comment. We made the suggested corrections in the text.

C12

C12

L-254: When referring to figure numbers, usually in the order in which they appear in the text, they should be read "(see Fig. 4 and 5)". If Figure 4 is mentioned before Figure 5 in the text, the order should be correct. If not, adjust the order of the numbers.

R12

Thank you for your comment. We made the suggested corrections in the text.

C13

C13

L-266-267: “In such a case, we obtain two reconstructions, which could increase the readability of the information. " Because "readability" is often used to describe the readability of text. Here, you can change to "clarity".

R13

Thank you for your comment. We made the suggested corrections in the text.

C14

C14

L-274: "M mask" may refer to a mask used to focus on a specific area, please give a detailed description and definition of the "M mask" in the context.

R14

Thank you for your comment. We have edited the paragraph as follows:

Old version:

Based on the results of Experiment II, we hypothesize that the algorithm also has the potential to be used for removing the source of noise when the signal associated with very soft tissues is treated as noise, e.g., by covering the entire object with the M mask, excluding backgrounds.

New version:

Based on the results of Experiment II, we hypothesize that the algorithm also has the potential to be used for removing the source of noise when the signal associated with very soft tissues is treated as noise, e.g., by determining the mask dividing image (projection) to object and background.

C15

C15

In the Discussion part, the results were not fully discussed, including the comparison with other methods, the limitations of the method and the direction of future improvement. Therefore, it is recommended to join the discussion.

R15

We have compared our method with the following methods: NMAR, LI-MAR and BHC Streaking developed by MItos.de and delivered with X-Aid software.

The results of this comparison are presented in the table, as well as representative slices and profiles. We have also added a new paragraph in the discussion with comments on the results of this comparison.

We have also identified the limitations of our method. A new paragraph has been added to the Discussion section:

The main limitation of our method is the strong dependence on the mask of the artifact source. That implies we must know the location of the artifact source before the reconstruction process. We also need to determine the artifact source's mask for each projection, which can be time-consuming.  The solution to this issue is to reconstruct the image without MAR, then create the mask of the artifact source, and then re-create the masking projections. However, it can still be hard to create the proper mask because of the metal artifacts.

Next, the specific feature of our method is the source of the artifact is completely removed from the image, not only artifacts. However, if the reconstruction of the metallic object is still needed, we can use the "hard tissue projection" for reconstruction.